# Exploring the role of routine brain imaging in detecting asymptomatic brain metastases in first diagnosed HER2-positive breast cancer

Carmen Leser[1]*, Fiona Kabashi[1], Georg Dorffner[2], Olgica Zaric[3,4], Christine Deutschmann[1], Daphne Gschwantler-Kaulich[1]

1 Department of Obstetrics and Gynecology, Medical University of Vienna, Vienna, Austria, 2 Institute for Artificial Intelligence, Medical University of Vienna, Vienna, Austria, 3 Research Centre for Medical Imaging and Image Analysis (MIAAI), Danube Private University, Krems, Austria, 4 Institute for Musculoskeletal imaging, Karl Landsteiner Society, St. Pölten, Austria

* carmen.leser@meduniwien.ac.at

## Abstract

### Background

HER2-positive breast cancer is leading to aggressive tumor growth and a higher risk of metastasis, particularly to the central nervous system (CNS). Routine brain imaging for asymptomatic HER2-positive patients is debated, with no current consensus; Given the severe clinical implications of brain metastases, further research is needed to determine the cost-effectiveness and clinical utility of routine imaging for high-risk patients to improve outcomes and inform targeted screening protocols.

### Methods

This retrospective, monocentric study at the General Hospital of Vienna (AKH Wien) examined female HER2-positive breast cancer patients at first diagnosis to assess brain metastasis from January 2019 to February 2024. The study included patients with asymptomatic confirmed HER2 positive breast cancer. Data were collected through comprehensive medical records and brain imaging with MRI.

### Results

Among 110 female patients meeting the inclusion criteria, 4 (3.6%) were diagnosed with brain metastases. Ki67 showed a marginal association with brain metastasis (p = 0.054), and tumor grade was a significant predictor, with intermediate differentiated tumors (G2 vs. G3) associated with a higher risk of brain metastases (p = 0.041) and brain metases are correlating with the axillary lymphnode status and the tumor sizeAlso, the absence of positive Östrogen and Progesteron receptors is a predictor in upcoming brain metastases (p < 0.001). Other factors like age were not significantly associated.

**Data availability statement:** All relevant data are within the manuscript and its Supporting Information files.

**Funding:** The author(s) received no specific funding for this work.

**Competing interests:** The authors have declared that no competing interests exist.

## Conclusion

This study found limited benefit in routine MRI for detecting asymptomatic brain metastases in HER2-positive breast cancer, given the low prevalence (3.6%). A targeted imaging approach for high-risk patients, like those with the absence of Hormon receptors and higher stage tumors, may be effective.

## Introduction

HER2 (Human Epidermal Growth Factor Receptor 2) is a gene that plays a critical role in the regulation of cell growth. In approximately 15–20% of breast cancer cases, the HER2 gene is amplified, leading to the overexpression of the HER2 protein on the surface of the tumor cells. This overexpression again is responsible for an aggressive tumor growth, resulting in poorer prognosis and an increase of the risk for metastasis when compared to other breast cancer subtypes, such as hormon receptor-positive (HR+) or HER2-negative [1]. Studies suggest that between 30–50% of patients with advanced HER2-positive breast-cancer will develop metastases to the central nervous system (CNS) at some point during the disease course [2–4]. One large study that followed 2686 breast cancer patients found that HER2-positive patients had a 10.1% rate of brain metastases, as opposed to 5.1% for HER2-negative patients [1]. The tendency of HER2-positive cancer to metastasize to the brain is largely due to the molecular characteristics of HER2 amplification, which enhances tumor cell invasiveness and ability to cross the blood-brain barrier (BBB) [5]. The development of brain metastasis has important clinical implications. Brain metastases not only significantly affect survival but also impact the patient's quality of life due to neurological symptoms, including headaches, seizures, cognitive impairment, and motor dysfunction. For many patients, brain metastasis is a major factor that limits overall survival, with many succumbing to brain progression rather than other systemic complications of the disease [6]. Thus, given the significant impact of brain metastasis, it should be evaluated whether routine brain imaging should be implemented for asymptomatic HER2-positive patients. Currently, there is no consensus on this. Magnetic resonance imaging (MRI) and positron emission tomography combined with computed tomography (PET-CT) are valuable tools that can detect brain metastasis early, particularly in asymptomatic cases. MRI has become the gold standard for detecting brain metastases due to its superior sensitivity in detecting small and asymptomatic lesions, especially for HER2-postive patients [7]. PET-CT imaging also plays a role, particularly in systemic disease assessment, but its use for brain imaging is less well-established due to the lower sensitivity of PET for brain lesions. Nonetheless, newer radiotracers and techniques are being developed to improve the utility of PET-CT in identifying brain metastases in HER2-positive patients [8]. Despite the risk of asymptomatic brain metastases in HER2-positive patients, routine screening is typically reserved for symptomatic patients. Meanwhile, the introduction of novel HER2-targeted therapies has changed the prognosis for patients

with HER2-positive breast cancer, including those with brain metastases. Agents such as tucatinib and trastuzumab-deruxtecan have shown promising results for patients with brain metastasis by demonstrating activity across the blood-brain barrier. Tucatinib, in combination with trastuzumab and capecitabine, has been shown to significantly reduce the risk of death and intracranial progression in this context [9]. Thus, an early detection through routine imaging could identify brain metastases when they are smaller and more treatable, potentially improving survival and reducing neuro-logical symptoms. This is why a recent review emphasized that the decision to perform brain imaging should consider the individual patient's risk factors, including HER2 status, tumor size, and previous metastatic spread [10]. Routine imaging, especially with advanced tools like MRI, might offer a survival benefit in high-risk patients, but evidence on cost-effectiveness and the number-needed-to-treat remains limited.

## Materials and methods

This retrospective, monocentric study was conducted at the Department of Gynecology, General Hospital of Vienna (AKH Wien), from January 2019 to February 2024. It employed a cross-sectional design, focusing on female HER2-positive breast cancer patients, particularly regarding the diagnosis and treatment of brain metastasis.

### Study population

Inclusion criteria were female patients aged 18 years or older, presenting for the first time at AKH Wien, with sufficient clinical documentation and HER2+++ or HER2++ confirmed by positive FISH testing. Exclusion criteria included male patients because we only treat women, patients under 18, HER2-negative breast cancer, pre-existing brain abnormalities, and those with ductal carcinoma in situ (DCIS) only.

### Data collection

Patients underwent medical history assessments, physical examinations, brain imaging with MRI at diagnosis, and histopathological evaluations. Asymptomatic patients were examined on a 3T MRI scanner (MAGNETON Vida, Siemens Healthineers, Erlangen, Germany) using 64-channer head-neck coil (Siemens, Erlangen, Germany) in supine posi-tion. Imaging protocol included T2w imaging with Sampling Perfection with Application optimized Contrast (t2_space) sequence, T2w with turbo spin echo (t2_tse) sequence, diffusion weighted imaging (DWI) with Readout Segmentation of Long Variable Echo-trains (RESOLVE), susceptibility weighted imaging (PWI), and contrast enhanced T1w imaging with high-resolution 3D isotropic Magnetization-Prepared Rapid Gradient-Echo (MPRAGE) sequence. Sequence parameters are provided in Table 1.

Data were collected from the RDA and AKIM systems from 10th September 2024–15th October 2024, pseudonymized, and entered into a Microsoft® Excel® file before being analyzed with IBM SPSS® Statistics software. Because of the

**Table 1. MRI sequences and parameters used for examinations of patients in this study.**

| MRI sequence | Acquisition type | Echo/Repetition/Inversion Time (TE/TR/IT) [ms] | In-plane resolution [mm²] | Slice thickness [mm] | Flip angle (FA) [°] | Bandwidth (BW) [Hz/px] |
|---|---|---|---|---|---|---|
| T2_space | 3D | 5000/398/1800 | 0.94 × 0.94 | 0.94 | 120 | 781 |
| T2_tse | 2D | 4870/89 | 0.57 × 0.57 | 3.0 | 150 | 250 |
| RESOLVE | 2D | 4190/86 | 0.43 × 0.43 | 4.0 | 180 | 574 |
| T2_swi | 3D | 28/20 | 0.90 × 0.90 | 1.75 | 15 | 120 |
| T1_MPRAGE | 3D | 2000/3.3/1010 | 1.0 × 1.0 | 1.0 | 9 | 250 |

*2D-two dimensional; 3D-three dimensional.

retrospective study, there was no need for participant content. Authors had access to identification of participants during data collection, but not after.

## Ethical considerations

The study was approved by the Ethics Committee of the Medical University of Vienna (EK-Nr. 1334/2024). Due to the retrospective design, no direct patient benefits or risks were anticipated, and data confidentiality was ensured via pseudonymization and secure data handling in compliance with good scientific practice.

In the present study, there are no conflicts of interest to disclose. All procedures performed involving human participants adhered to the ethical standards of the institutional research committee and the 1964 Helsinki Declaration and its later amendments. Informed consent was obtained from all individual participants included in the study.

This declaration aims to assure the research community and the public of our commitment to ethical conduct and integrity in our research. We understand the importance of these ethical considerations and believe that our adherence to these principles will contribute to the credibility and reliability of our findings.

## Statistical analysis

Statistical analyses were conducted using IBM SPSS® Statistics, version 22. Descriptive statistics, including mean (M), standard deviation (SD), median (Md), and interquartile range (IQR), were used for continuous variables, with categorical variables summarized by frequencies (n) and percentages (%). For group comparisons, the non-parametric Mann-Whitney U-test was used, while chi-square ($\chi^2$) tests or Fisher's exact test were applied for categorical variables. Significance was set at $p \leq 0.05$.

## Results

A total of 110 female patients meeting the inclusion criteria, were included in the analysis. The mean age of the patients was 53.7 years (SD = 13.0 years), and the mean body mass index (BMI) was 25.8 kg/m² (SD = 6.5 kg/m²).

The Ki67 proliferation index showed a mean of 46.01% (SD = 20.7%). A total of 91 patients (82.7%) had a Ki67 index above the cutoff value of 25%, indicating a high level of cellular proliferation. No significant correlation was found between Ki67 levels and patient age (p = 0.637), indicating that age did not influence the proliferation index. The most common tumor type in the study population was invasive ductal carcinoma (IDC), present in 91.8% of cases (n = 101),. A small percentage of patients had invasive lobular carcinoma (ILC) (4.5%) or other types of cancer, such as mucinous or micropapillary carcinoma (5.5%) Additionally, one patient exhibiting a rare combination of IDC, and another tumor type. Tumor size, an essential part of the staging process, was documented for 110 patients. The majority of tumors were classified in TNM as clinial tumor size(cT)1 (n = 49, 44.5%), while 32 (29.1%) were classified cT 2, 9(8.2%) were classified cT3, and 19 (17.3%) were classified as cT4. While the tumor size in the patients with brain metastases differ (1 out of 49 patients cT1, 1 out of 9 patients cT3, 2 out of 19 patients cT4), all patients had positive axillary lymphnodes. This is a significant finding (p < 0.001). Only one of them had also metastases in other organs than the brain (bone metastases). 13 patients suffered from metastases other than brain.

Brain metastases were diagnosed in 4 out of the 110 patients (3.6%). The brain metastases were all just found with imaging and were not proofed with a biopsy. 2 suffered from a single brain metastases with a size small than 15 mm. One patient showed 2 brain metastases with the biggest size of 25 mm and one patient was found with 4 lesions in the brain with the biggest size of 25 mm. A chi-square test was conducted to examine the association between a family history of cancer and the presence of brain metastases. However, the analysis did not reveal a significant association, p > 0.999. Furthermore, a chi-square test was performed to investigate the relationship between MRI performance and the detection of brain metastases.

Estrogen receptor (ER) and progesterone receptor (PR) status were tested as part of the initial diagnostic workup. The majority of patients (n = 68, 61.8%) were ER-positive, while 52 (47.3%) were PR-positive. Among the cohort, 46.4%

of patients were positive for both receptors, while 37.3% were negative for both receptors. 3 out of 4 patients with brain metastases were Hormon receptor negative. So, 1 out of 69 patients with a Hormon receptor positive Carcinoma and 3 out of 41 Hormon receptor negative breast cancer patients suffered from brain metastases. This is a significant finding (p < 0.001) (Fig 1).

The most commonly prescribed regimen was EC THP (epirubicin and cyclophosphamide, followed by docetaxel, trastuzumab, and pertuzumab), given to 67.3% of patients. Other regimens, such as carboplatin or TDM1, were less frequently administered. Among the four patients diagnosed with brain metastases, three underwent subsequent treatment involving either Gamma Knife radiosurgery or whole-brain radiation therapy. Two patients initially received EC THP, while the other two were treated with THP alone.

A logistic regression model was applied to assess the predictors of brain metastases, including age, Ki67 index, tumor grade, and tumor type. (Fig 2) Age did not significantly predict the presence of brain metastases (p > 0.05). However, Ki67 showed a marginal association with brain metastasis in one model (p = 0.054), and tumor grade was a significant predictor, with intermediate differentiated tumors (G2 vs. G3) associated with a higher risk of brain metastases (p = 0.041). We found 1 patient with brain metastases in 68 G3 tumors and 3 out of 40 G2 tumors.

## Discussion

This study aimed to assess the utility of routine MRI in detecting asymptomatic brain metastases in HER2-positive breast cancer patients. Findings revealed a low prevalence of asymptomatic brain metastases, with only 4 of 110 patients (3.6%) diagnosed through these imaging methods. This aligns with previous research indicating that brain metastases are rarely detected in early, asymptomatic stages of HER2-positive breast cancer. For instance, Kaplan et al. (2013) found that brain metastases were infrequently observed in asymptomatic, non-metastatic HER2-positive patients, suggesting limited benefit from routine cranial imaging in this group [11] and finding no prolonged survival rate, when detecting metastasis earlier. But since this study is more than 10 years old, the question is, how the outcome would be with new therapies like Trastuzumab Deruxtecan. Maurer et al. (2018) reported that routine MRI in asymptomatic patients showed a better overall

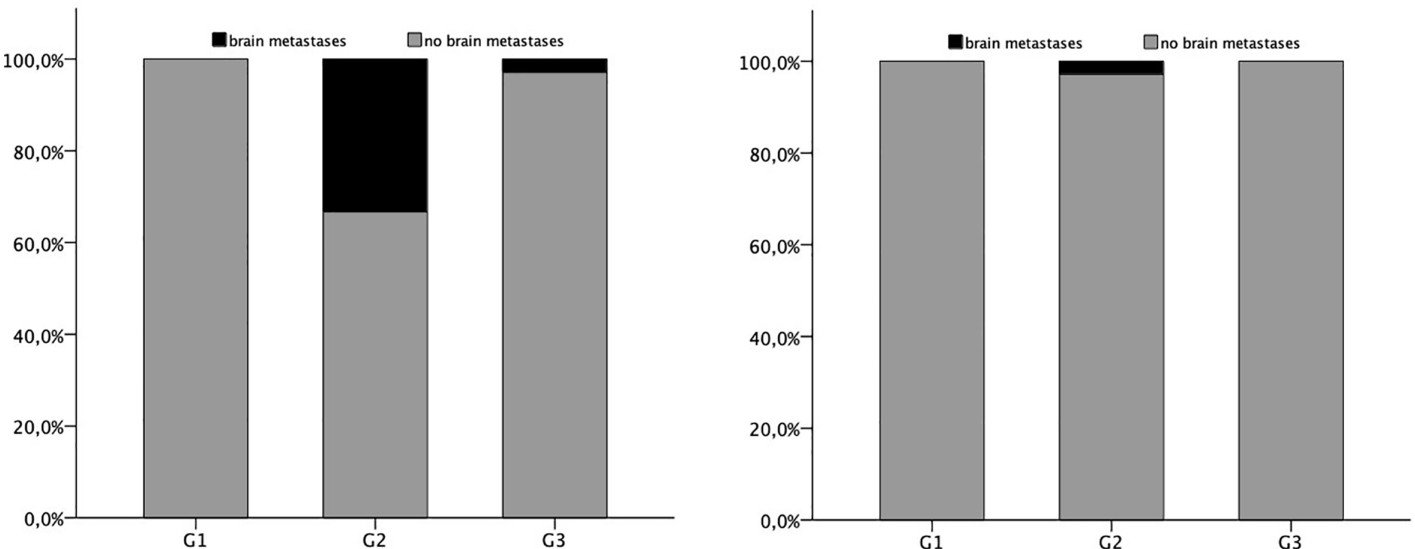

**Fig 1.  a: Brain metastasis in hormonreceptor negative patients.** Proportions of brain metastasis occurred in estrogen and progesterone negative patients compared with grading. b: Brain metastasis in hormonreceptor positive patients. Proportions of brain metastasis occurred in estrogen and/or progesterone positive patients compared with grading.

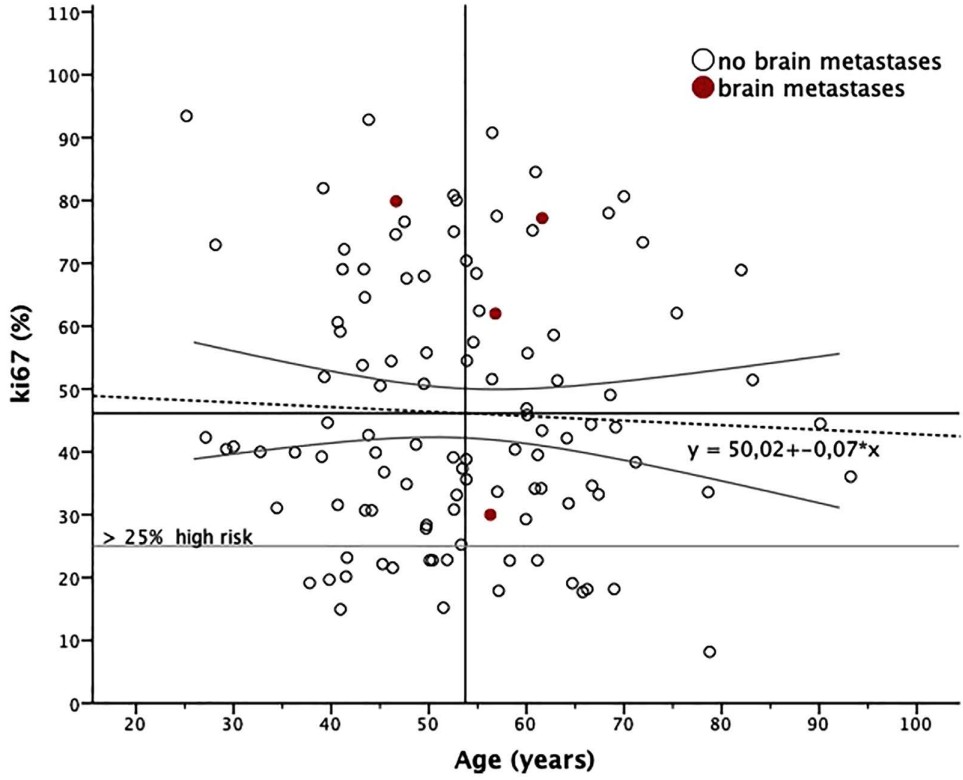

**Fig 2. Relation between age and ki67.** Bivariate scatter plot for the relationship between age (x-axis, in years) and the cell division marker (y-axis) with linear regression function (95% CI) in n = 110 patients.

survival than symptomatic patients [12]. The low detection rate in this study suggests that, while HER2-positive status is a known risk factor, asymptomatic early-stage patients seldom present with detectable brain lesions. Instead, brain metastases tend to appear later, often with significant morbidity and rapid disease progression. This finding challenges the clinical value of routine brain imaging for all asymptomatic HER2-positive patients. Although MRI is an effective tool for imaging brain lesions, their limited sensitivity for early, asymptomatic metastases—especially in patients without additional risk factors—supports a more selective screening approach [4].

**Ki67 index and tumor characteristics**

In our study, we wanted to understand the relationship between age, Ki67 proliferation index, tumor grade, and the risk of brain metastasis in HER2-positive breast cancer patients. Our findings indicated that age alone did not correlate significantly with Ki67 levels, nor did it serve as a predictor for brain metastasis in our logistic regression model. The absence of a significant correlation between age and Ki67 aligns with previous studies, which have also found that Ki67—though associated with tumor aggressiveness—does not consistently correlate with patient age in breast cancer cohorts [16]. This suggests that Ki67 as a marker of proliferation reflects tumor biology more directly than demographic characteristics like age. However, while age did not show any significant association with brain metastasis, Ki67 did present a marginal association with metastasis risk (p = 0.072). This aligns with findings from Song et al. (2019), who noted that high Ki67 levels are often indicative of more aggressive tumor behavior, albeit not as a standalone predictor for metastatic spread [17]. Due to the predictive limitations of Ki67 for metastasis, it needs to be considered a part of a broader panel of markers rather than an independent prognostic tool. On the other hand, tumor grade emerged as a significant predictor for brain

metastasis, with intermediate-grade tumors (G2 vs. G3) showing a notably increased risk of brain metastasis (p = 0.041). This correlation is inconsistent with literature suggesting that higher histological grades in HER2-positive breast cancer are associated with a greater likelihood of central nervous system (CNS) involvement [18,19]. According to studies, we found significant predictors like tumor size and axillary lymphnode status [20,21].

### Hormon receptor status and brain metastasis

Our study found that while most patients in our cohort were estrogen receptor (ER)-positive (60.9%) and progesterone receptor (PR)-positive (47.2%), those patients without positive hormon receptors were more likely to suffer from brain metastases. This aligns with previous research that shows hormon receptor-negative, HER2-positive breast cancers often exhibit a higher risk of brain metastasis due to their aggressive nature. However, this tendency for brain metastasis does not usually appear in the early, symptom-free stages of HER2-positive breast cancer. [22,23]. A study by Garcia-Alvarez et al. (2021) indicated that while hormon receptor-negative, HER2-positive cancers are overrepresented among patients with brain metastases, brain metastases can still develop in ER-positive, HER2-positive tumors, likely influenced by other biological and molecular factors [6]. This points to a more nuanced interplay between hormon receptor status and HER2 overexpression. Furthermore, receptor discordance—where hormon receptor status shifts between the primary tumor and metastatic sites – has been documented. Sperduto et al. (2020) found that up to 25–37.5% of patients experience a loss of hormon receptor expression in brain metastases, suggesting that these tumors may become more aggressive as they progress, particularly in HER2-positive cases [24]. This evidence supports the potential value of reassessing receptor status in metastatic disease stages, as conversion from hormon receptor-positive to negative in metastatic lesions may drive a shift in treatment strategies.

### Implications for routine imaging in asymptomatic HER2-positive patients

Elaborating on routine MRI for asymptomatic HER2-positive breast cancer patients, the findings from our study suggest that a broad imaging approach may be unwarranted due to the low incidence of brain metastases identified in our cohort (3.6%). The cost-effectiveness and clinical value of such routine imaging have thus been questioned, as indiscriminate imaging may lead to unnecessary costs and heightened patient anxiety without significant clinical benefit [25]. Furthermore, advanced HER2-targeted imaging techniques, such as $^{68}Ga$-ABY-025 PET/CT and $^{64}Cu$-DOTA-trastuzumab PET, have demonstrated higher specificity for detecting HER2-positive brain lesions. These tracers may allow for more selective screening of HER2-positive metastases and offer a potential alternative to routine imaging in low-risk cases [13,14]. Furthermore, some research proposes alternative, symptom-driven or risk-adjusted screening protocols, where patients with a higher likelihood of brain metastasis development, such as those with additional metastatic sites or certain molecular markers, might benefit more from regular imaging [15].

Our low yield of positive imaging findings rather underscores the potential benefit of refining imaging protocols to focus on patients with additional risk factors. Specifically, targeted imaging approaches like PET with HER2-specific tracers may improve early detection and reduce unnecessary imaging in lower-risk patients, supporting a more efficient and clinically valuable strategy in managing HER2-positive breast cancer patients at risk for brain metastasis. Studies highlight that, while imaging techniques like PET/CT can reduce false positives and accelerate treatment in high-risk breast cancer cohorts, they are most cost-effective when targeted to specific patient groups rather than applied universally [26]. Moreover, the cost per positive finding is significant, prompting questions about the practicality of such routine applications. Cost-effectiveness studies further support selective imaging based on a patient's individual risk profile. A study on the economic value of PET/CT in breast cancer stages II-III highlights that PET/CT is only marginally cost-effective when applied in settings where the probability of metastasis is inherently low [27]. Such findings suggest that broad, indiscriminate imaging may contribute to overdiagnosis and overtreatment, especially when applied to early-stage, asymptomatic patients without clear indicators of metastatic disease. Thus, recent studies support a more selective approach. For

example, patients with additional risk factors, such as higher-grade tumors or metastatic spread to other high-risk sites, may benefit more from targeted imaging interventions [28]. Given that HER2-positive patients with advanced or recurrent disease show a propensity for brain metastases, additional monitoring might be prioritized in patients with these features.

Moreover, research on advanced HER2-targeted imaging modalities, such as the ^68Ga-ABY-025 PET/CT, demonstrates a promising ability to selectively identify HER2-positive metastases with greater specificity than conventional imaging techniques. The small peptide tracer ABY-025, designed to bind uniquely to the HER2 receptor without competing with therapeutic antibodies, has shown potential for distinguishing HER2-positive from HER2-negative lesions, even in complex metastatic contexts [7]. In clinical studies, ^68Ga-ABY-025 PET/CT effectively quantified HER2 expression across whole-body scans, capturing dynamic changes in HER2 expression across different metastatic sites [29]. Additionally, findings suggest that ^68Ga-ABY-025 PET/CT could serve as an efficient alternative to broad routine imaging, particularly in HER2-positive breast cancer patients where routine MRI screening may yield limited value due to the low incidence of asymptomatic brain metastases [30]. This approach could help reduce unnecessary imaging, minimizing patient exposure to radiation and the likelihood of overdiagnosis.

Given these promising results, future studies on ^68Ga-ABY-025 could explore its use as a routine diagnostic tool within targeted surveillance protocols. This would involve further assessing its sensitivity and specificity across larger patient populations and expanding on cost-effectiveness analyses to clarify its value relative to traditional imaging approaches.

### Limitations and future directions

This study has limitations that may impact the interpretation and generalizability of our findings. Firstly, the retrospective design introduces limitations related to data availability and accuracy, as well as potential biases from incomplete records, which could affect findings on factors associated with brain metastasis. A prospective design would enable more controlled data collection and better control for confounding factors. Additionally, our relatively small sample size (n = 110) limits statistical power, making it difficult to fully explore associations within specific patient subgroups. Moreover, our study did not include advanced molecular markers or genetic profiling, which could help identify higher-risk patients more precisely, supporting more targeted imaging approaches. We also did not evaluate newer imaging methods, such as HER2-specific PET tracers, which may offer greater specificity in detecting brain metastases in HER2-positive patients. Perhaps in the future genetic molecular profiling and artificial intelligence in breast cancer diagnosis and imaging of metastases can lead to new perspectives [31].

### Conclusion

The present study indicates that routine MRI may offer limited benefit for detecting asymptomatic brain metastases in HER2-positive breast cancer patients due to the low prevalence and limited predictive factors observed. A more targeted imaging strategy focused on high-risk patients, such as patients with the absence of Hormon receptors and higher stage tumors, could improve cost-efficiency and clinical relevance. Future research should also explore advanced imaging modalities, like HER2-specific PET tracers, to improve specificity in detecting HER2-positive metastases.

### Supporting information

**S1 File. Final data set as excel sheet.**
(XLSX)

### Author contributions

**Conceptualization:** Carmen Leser, Daphne Gschwantler-Kaulich.

**Data curation:** Carmen Leser, Fiona Kabashi, Christine Deutschmann.

**Formal analysis:** Fiona Kabashi, Georg Dorffner.

**Methodology:** Carmen Leser.

**Supervision:** Daphne Gschwantler-Kaulich.

**Visualization:** Fiona Kabashi.

**Writing – original draft:** Carmen Leser.

**Writing – review & editing:** Olgica Zaric, Christine Deutschmann, Daphne Gschwantler-Kaulich.

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
