## [Decision Letter · Decision Letter 0]

2 Jan 2025

Dear Dr. Leser, 

Thank you for submitting your manuscript to PLOS ONE. After careful consideration, we feel that it has merit but does not fully meet PLOS ONE’s publication criteria as it currently stands. Therefore, we invite you to submit a revised version of the manuscript that addresses the points raised during the review process.

We look forward to receiving your revised manuscript.

Kind regards,

Elingarami Sauli, PhD

Academic Editor

PLOS ONE

Journal Requirements:

3. We note that your Data Availability Statement is currently as follows: 

“All relevant data are within the manuscript and its Supporting Information files.”

**Additional Editor Comments:**

When responding to reviewers comments please make sure to detail your imaging protocols, with respective justification of their use in your research study.

Reviewers' comments:

Reviewer's Responses to Questions

**Comments to the Author**

1. Is the manuscript technically sound, and do the data support the conclusions?

Reviewer #1: Partly

Reviewer #2: Partly

2. Has the statistical analysis been performed appropriately and rigorously?

Reviewer #1: Yes

Reviewer #2: No

3. Have the authors made all data underlying the findings in their manuscript fully available?

Reviewer #1: Yes

Reviewer #2: No

4. Is the manuscript presented in an intelligible fashion and written in standard English?

Reviewer #1: Yes

Reviewer #2: Yes

Reviewer #1: General Comments:

In this manuscript, the authors evaluate the utility of routine PET-CT and MRI for detecting asymptomatic brain metastases in HER2-positive breast cancer patients. The study addresses a clinically significant issue and provides valuable insights into the limited role of routine imaging in this population. The manuscript highlights the low detection rate of brain metastases and discusses alternative imaging strategies, including advanced HER2-specific modalities. This is an important contribution to the field of oncology imaging, particularly in optimizing cost-effective and targeted screening protocols.

However, there are several areas that require major revisions to improve the clarity, methodological transparency, and overall impact of the manuscript. Key concerns include the lack of detailed imaging protocols (e.g., MRI acquisition techniques), the justification for using PET-CT in the assessment of brain metastases, and a more robust discussion of limitations and implications for clinical practice. Addressing these issues is essential to strengthen the manuscript's scientific rigor and clinical relevance.

Specific comments:

1. Comment on the Use of PET-CT

In the Methods and Results sections, it is mentioned that PET-CT was used for the evaluation of brain metastases. However, FDG PET-CT is known to be inferior to contrast-enhanced MRI in detecting metastatic brain tumors (1) and generally not suitable for detecting brain tumors or metastatic brain lesions. Please clarify the purpose of using PET-CT in this study and how its findings are interpreted. Additionally, elaborate on the impact of PET-CT results on the overall conclusions of the study.

2. Comment on the Imaging Protocol for Contrast-Enhanced MRI

In the Results section, it is stated that brain metastases were detected using MRI; however, the specific imaging protocol for contrast-enhanced MRI is not provided. Studies have shown that 3D contrast-enhanced MRI is superior to 2D methods for detecting metastatic brain lesions (2). Furthermore, the ability to detect metastatic brain tumors in contrast-enhanced MRI scans depends on the imaging device (1.5T vs. 3T, etc.), the dose of contrast agent, and other factors (3) (4). Providing detailed information about the MRI protocol (e.g., 2D or 3D imaging, slice thickness, etc.) would enhance the reliability and interpretability of the results .

3. Suggestion to Limit the Analysis to MRI-Only Patients

For the diagnosis of metastatic brain lesions, it would be more consistent and reliable to limit the analysis to patients who underwent MRI, rather than including those evaluated with PET-CT. I recommend reconsidering the inclusion criteria and presenting results based solely on MRI findings.

1. Krüger S, Mottaghy FM, Buck AK, Maschke S, Kley H, Frechen D, et al. Brain metastasis in lung cancer. Comparison of cerebral MRI and 18F-FDG-PET/CT for diagnosis in the initial staging. Nuklearmedizin. 2011;50(3):101-6.

2. Kakeda S, Korogi Y, Hiai Y, Ohnari N, Moriya J, Kamada K, et al. Detection of brain metastasis at 3T: comparison among SE, IR-FSE and 3D-GRE sequences. European Radiology. 2007;17(9):2345-51.

3. Ochi T, Taoka T, Matsuda R, Sakamoto M, Akashi T, Tamamoto T, et al. Comparison between Two Separate Injections and a Single Injection of Double-dose Contrast Medium for Contrast-enhanced MR Imaging of Metastatic Brain Tumors. Magnetic Resonance in Medical Sciences. 2014;13(4):221-9.

4. Jensen LG, Nath SK, Murphy KT, Alksne JF, Mundt AJ, Lawson JD. Difference in Number of Brain Metastases Detected using 1.5T and 3T MRI in Patients Treated with Stereotactic Radiosurgery. International Journal of Radiation Oncology, Biology, Physics. 2011;81(2):S300.

Reviewer #2: In this article, Leser et al. evaluated the relevant topic of proactive brain imaging in patients with HER2-positive breast cancer. The authors analyzed a cohort of 153 patients with newly diagnosed HER2-positive breast cancer, identifying only 4 patients with brain metastases. However, several significant limitations render this study unsuitable for publication:

-Patient Cohort Selection: The authors included patients at any disease stage, rather than focusing on those with advanced-stage disease, which is the clinical setting where brain imaging is typically considered, undermining the relevance of the findings. Furthermore, the disease stage of the patients with brain metastases is not reported.

-Screening Methodology: It appears that PET/CT scans were used for brain metastasis screening in some cases, which constitutes a major methodological flaw. As the authors themselves acknowledge, PET/CT is inadequate for detecting brain metastases. Indeed, all identified brain metastases were diagnosed via MRI, further highlighting the inappropriate reliance on PET/CT.

-Small Sample Size: The extremely low number of cases with brain metastases (n=4) makes the associations reported in the study highly unstable. Additionally, disease stage—an expectedly strong predictor of brain involvement—could heavily confound any observed associations.

**Do you want your identity to be public for this peer review?** For information about this choice, including consent withdrawal, please see our Privacy Policy

Reviewer #1: No

Reviewer #2: No

---

## [Author Response · Author response to Decision Letter 1]

16 Jan 2025

Dear Reviewers,

Thank you for your great comments on our manuscript. We worked hard to revise the manuscript, and changed the whole statistics to include patients only with a brain MRI.

Reviewer #1: General Comments:

In this manuscript, the authors evaluate the utility of routine PET-CT and MRI for detecting asymptomatic brain metastases in HER2-positive breast cancer patients. The study addresses a clinically significant issue and provides valuable insights into the limited role of routine imaging in this population. The manuscript highlights the low detection rate of brain metastases and discusses alternative imaging strategies, including advanced HER2-specific modalities. This is an important contribution to the field of oncology imaging, particularly in optimizing cost-effective and targeted screening protocols.

However, there are several areas that require major revisions to improve the clarity, methodological transparency, and overall impact of the manuscript. Key concerns include the lack of detailed imaging protocols (e.g., MRI acquisition techniques), the justification for using PET-CT in the assessment of brain metastases, and a more robust discussion of limitations and implications for clinical practice. Addressing these issues is essential to strengthen the manuscript's scientific rigor and clinical relevance.

Specific comments:

1. Comment on the Use of PET-CT

In the Methods and Results sections, it is mentioned that PET-CT was used for the evaluation of brain metastases. However, FDG PET-CT is known to be inferior to contrast-enhanced MRI in detecting metastatic brain tumors (1) and generally not suitable for detecting brain tumors or metastatic brain lesions. Please clarify the purpose of using PET-CT in this study and how its findings are interpreted. Additionally, elaborate on the impact of PET-CT results on the overall conclusions of the study. Thanks for your comments. We strongly thought about it and decided to follow your suggestion in point 3.

2. Comment on the Imaging Protocol for Contrast-Enhanced MRI

In the Results section, it is stated that brain metastases were detected using MRI; however, the specific imaging protocol for contrast-enhanced MRI is not provided. Studies have shown that 3D contrast-enhanced MRI is superior to 2D methods for detecting metastatic brain lesions (2). Furthermore, the ability to detect metastatic brain tumors in contrast-enhanced MRI scans depends on the imaging device (1.5T vs. 3T, etc.), the dose of contrast agent, and other factors (3) (4). Providing detailed information about the MRI protocol (e.g., 2D or 3D imaging, slice thickness, etc.) would enhance the reliability and interpretability of the results. Thanks for the comment. We tried to add all the missing stuff.

3. Suggestion to Limit the Analysis to MRI-Only Patients

For the diagnosis of metastatic brain lesions, it would be more consistent and reliable to limit the analysis to patients who underwent MRI, rather than including those evaluated with PET-CT. I recommend reconsidering the inclusion criteria and presenting results based solely on MRI findings. We followed your suggestion and are happy, that this recreation made the paper to a better one. Thanks again for your thoughtful comments.

1. Krüger S, Mottaghy FM, Buck AK, Maschke S, Kley H, Frechen D, et al. Brain metastasis in lung cancer. Comparison of cerebral MRI and 18F-FDG-PET/CT for diagnosis in the initial staging. Nuklearmedizin. 2011;50(3):101-6.

2. Kakeda S, Korogi Y, Hiai Y, Ohnari N, Moriya J, Kamada K, et al. Detection of brain metastasis at 3T: comparison among SE, IR-FSE and 3D-GRE sequences. European Radiology. 2007;17(9):2345-51.

3. Ochi T, Taoka T, Matsuda R, Sakamoto M, Akashi T, Tamamoto T, et al. Comparison between Two Separate Injections and a Single Injection of Double-dose Contrast Medium for Contrast-enhanced MR Imaging of Metastatic Brain Tumors. Magnetic Resonance in Medical Sciences. 2014;13(4):221-9.

4. Jensen LG, Nath SK, Murphy KT, Alksne JF, Mundt AJ, Lawson JD. Difference in Number of Brain Metastases Detected using 1.5T and 3T MRI in Patients Treated with Stereotactic Radiosurgery. International Journal of Radiation Oncology, Biology, Physics. 2011;81(2):S300.

Reviewer #2: In this article, Leser et al. evaluated the relevant topic of proactive brain imaging in patients with HER2-positive breast cancer. The authors analyzed a cohort of 153 patients with newly diagnosed HER2-positive breast cancer, identifying only 4 patients with brain metastases. However, several significant limitations render this study unsuitable for publication:

-Patient Cohort Selection: The authors included patients at any disease stage, rather than focusing on those with advanced-stage disease, which is the clinical setting where brain imaging is typically considered, undermining the relevance of the findings. Furthermore, the disease stage of the patients with brain metastases is not reported. We added the stages of the patients with brain metastases and will follow your suggestion in further prospective studies. Thank you for the suggestion.

-Screening Methodology: It appears that PET/CT scans were used for brain metastasis screening in some cases, which constitutes a major methodological flaw. As the authors themselves acknowledge, PET/CT is inadequate for detecting brain metastases. Indeed, all identified brain metastases were diagnosed via MRI, further highlighting the inappropriate reliance on PET/CT. Thanks for your comment. Because of your suggestion and the one from the other reviewer we limit the analysis to MRI-only patients.

-Small Sample Size: The extremely low number of cases with brain metastases (n=4) makes the associations reported in the study highly unstable. Additionally, disease stage—an expectedly strong predictor of brain involvement—could heavily confound any observed associations. Thank you for your thoughtful comment. Because of your comment, we looked on the correlation between stage and lymphnode status and it was significant. So, thanks again to improve the results with your impact.

I hope that the changes meet your expectations, and are bringing our manuscript closer to publishing in your manuscript.

Greets

---

## [Decision Letter · Decision Letter 1]

10 Mar 2025

Dear Dr. Leser,

Thank you for submitting your manuscript to PLOS ONE. After careful consideration, we feel that it has merit but does not fully meet PLOS ONE’s publication criteria as it currently stands. Therefore, we invite you to submit a revised version of the manuscript that addresses the points raised during the review process.

We look forward to receiving your revised manuscript.

Kind regards,

Elingarami Sauli, PhD

Academic Editor

PLOS ONE

Journal Requirements:

Reviewers' comments:

Reviewer's Responses to Questions

**Comments to the Author**

Reviewer #3: All comments have been addressed

Reviewer #4: All comments have been addressed

Reviewer #5: All comments have been addressed

2. Is the manuscript technically sound, and do the data support the conclusions?

Reviewer #3: Yes

Reviewer #4: No

Reviewer #5: Yes

3. Has the statistical analysis been performed appropriately and rigorously?

Reviewer #3: Yes

Reviewer #4: Yes

Reviewer #5: Yes

4. Have the authors made all data underlying the findings in their manuscript fully available?

Reviewer #3: Yes

Reviewer #4: Yes

Reviewer #5: Yes

5. Is the manuscript presented in an intelligible fashion and written in standard English?

Reviewer #3: Yes

Reviewer #4: Yes

Reviewer #5: Yes

Reviewer #3: dear authors, I have no more comments, thanks for working on the revisions.dear authors, I have no more comments, thanks for working on the revisions.dear authors, I have no more comments, thanks for working on the revisions.dear authors, I have no more comments, thanks for working on the revisions.dear authors, I have no more comments, thanks for working on the revisions.dear authors, I have no more comments, thanks for working on the revisions.

Reviewer #4: There are some significant flaws to be addressed with the manuscript.

The patient population must be better defined, why were males excluded, did all these women undergo resection with curative intent, how many received neo-adjuvant systemic therapy (consider standard of care for women with Her2 positive disease).

Was there any evidence of extra-cranial metastatic disease

At what time point was the MRI done? at diagnosis? Were these women symptomatic? Describe the size and number of lesions that were detected in these 4 women.

Please define the tumor characteristics using TNM classification, for size and nodal status, ot explain the system used.

DCIS was an exclusion criteria (obviously as by definition these cannot have metastatic disease), yet we are told in great detail how many women had DCIS, not relevant.

The spelling or Oestrogen/estrogen hormon/hormone status needs to be consistent throughout the paper.

Unfortunately the very small sample size (4 brain tumours detected) makes this study of limited utility.

Reviewer #5: In this article, the author claimed the clinical value of routine brain imaging for all asymptomatic HER2-positive patients may be limited. The patients with a higher likelihood of brain metastasis development, such as those with additional metastatic sites or certain molecular markers, might benefit more from regular imaging. In addition, HER2-specific PET imaging may improve early detection and reduce unnecessary imaging in lower-risk patients, supporting a more efficient and clinically valuable strategy in managing HER2-positive breast cancer patients at risk for brain metastasis. This is a very interesting study. But there are still a few points that confuse me.

1.In the result, the author claimed that “ The majority of tumors were classified as size category 1 (n = 469, 445.54%), while 3248 (2931.16%) were classified category 2, 13 9(8.26%) were classified category 3, and 22 19 (174.35%) were classified as category 4. “ Please supplement the grading method in the methodology section.

2.Among the 110 patients included, how many patients actually had brain metastases, and was there any situation of missed diagnosis by MRI?

3.In Discussion section, the potential value of HER2-PET imaging application has been discussed in the text before and after the Discussion. Please merge them.

4.In Discussion section, The author extensively discusses “Ki67 Index and Tumor Characteristics”, “Hormone Receptor Status and Brain Metastasis”, “Implications for Routine Imaging in Asymptomatic HER2-Positive Patients”. It is suggested that the significance of referring to related molecular indicators to guide patients to choose brain MRI should be emphasized.

**Do you want your identity to be public for this peer review?** For information about this choice, including consent withdrawal, please see our Privacy Policy

Reviewer #3: No

Reviewer #4: No

Reviewer #5: No

---

## [Author Response · Author response to Decision Letter 2]

15 Apr 2025

Thanks for your comments. Hopefully we met your thoughts.

---

## [Decision Letter · Decision Letter 2]

1 Oct 2025

Exploring the Role of Routine Brain Imaging in Detecting Asymptomatic Brain Metastases in First Diagnosed HER2-Positive Breast Cancer

PONE-D-24-55756R2

Dear Dr. Carmen,

We’re pleased to inform you that your manuscript has been judged scientifically suitable for publication and will be formally accepted for publication once it meets all outstanding technical requirements.

Kind regards,

Elingarami Sauli, PhD

Academic Editor

PLOS ONE

Additional Editor Comments (optional):

This submission can now be accepted after incorporation of all key comments from reviewers by authors, including proper addressing of all observed limitations.

Reviewers' comments:

Reviewer's Responses to Questions

**Comments to the Author**

Reviewer #4: All comments have been addressed

Reviewer #5: All comments have been addressed

2. Is the manuscript technically sound, and do the data support the conclusions?

Reviewer #4: Yes

Reviewer #5: Yes

3. Has the statistical analysis been performed appropriately and rigorously?

Reviewer #4: Yes

Reviewer #5: Yes

4. Have the authors made all data underlying the findings in their manuscript fully available?

Reviewer #4: Yes

Reviewer #5: Yes

5. Is the manuscript presented in an intelligible fashion and written in standard English?

Reviewer #4: Yes

Reviewer #5: Yes

Reviewer #4: acceptable revisions made and concerns raised addressed in this current version

Reviewer #5: 1.In Result "The most common tumor type in the study population was invasive ductal carcinoma (IDC), present in 91.8% of cases (n = 101). A small percentage of patients had invasive lobular carcinoma (ILC) (4.5%) or other types of cancer, such as mucinous or micropapillary carcinoma (5.5%)”. The reported percentages for tumor type distribution do not sum to 100% (91.8% + 4.5% + 5.5% = 101.8%). Please verify and correct these values.

2.In Result, “Tumor size, an essential part of the staging process, was documented for 110 patients. The majority of tumors were classified in TNM as clinial tumor size(cT)size category 1 (n = 49, 44.5%), while 32 (29.1%) were classified category cT 2, 9(8.2%) were classified category cT3, and 19 (17.3%) were classified as cT4”. The total number of cases across cT categories sums to 109 (49+32+9+19) rather than the documented 110 patients. And please clarify this discrepancy."Please include in the Methods section the size criteria used for cT classification to improve reproducibility.

3.Comment on numerical formatting: When numbers begin a sentence, they should be written in word form (e.g., 3 out of 4 patients with brain metastases were Hormon receptor negative.'). Please revise all instances accordingly.

4.The manuscript contains several unclear or ambiguous statements. For example: "Only one of them had also metastases in other organs than the brain (bone metastases). 13 patients suffered from metastases other than brain." Suggested revision: Among the 110 patients, 17 (right?) had distant metastases, including 4 with brain metastases.

5.Additional suggestions:

For percentage reporting: "Consider presenting percentages with one decimal place consistently throughout the manuscript for uniformity."

For data verification: "Please double-check all numerical values in the manuscript for internal consistency."

**Do you want your identity to be public for this peer review?** For information about this choice, including consent withdrawal, please see our Privacy Policy

Reviewer #4: No

Reviewer #5: No

---

## [Editor Report · Acceptance letter]

PONE-D-24-55756R2

PLOS ONE

Dear Dr. Leser,

I'm pleased to inform you that your manuscript has been deemed suitable for publication in PLOS ONE. Congratulations! Your manuscript is now being handed over to our production team.

Kind regards,

on behalf of

Dr. Elingarami Sauli

Academic Editor

PLOS ONE